# Language Ranker: A Lightweight Ranking framework for LLM Decoding

**Chenheng Zhang**[1*]      **Tianqi Du**[1*]    **Jizhe Zhang**[1]    **Mingqing Xiao**[1,4]    **Yifei Wang**[3]
**Yisen Wang**[1,2†]      **Zhouchen Lin**[1,2†]

[1]State Key Lab of General Artificial Intelligence,
School of Intelligence Science and Technology, Peking University
[2]Institute for Artificial Intelligence, Peking University
[3]MIT CSAIL, MA, USA
[4]Microsoft Research Asia

## Abstract

Conventional research on large language models (LLMs) has primarily focused on refining output distributions, while paying less attention to the decoding process that transforms these distributions into final responses. Recent advances, such as scaling the computation of inference time with reward models, have underscored the importance of decoding, but these methods often suffer from high computational costs and limited applicability. In this paper, ***we revisit LLM generation through the lens of recommender systems***, conceptualizing the decoding process as analogous to the ranking stage in recommendation pipelines. From this perspective, we observe that both traditional decoding methods and reward models exhibit clear limitations such as redundancy. Motivated by this insight, we propose **Language Ranker**, a novel framework that introduces a lightweight module to rerank candidate responses using features extracted by the base model. Experiments across a wide range of tasks show that Language Ranker achieves performance comparable to large-scale reward models, while requiring only **<0.5M** additional parameters, significantly reducing the computational overhead during both training and inference stages. This highlights the efficiency and effectiveness of our method, showcasing its potential to fully unlock the capabilities of LLMs. The implementation is released at https://github.com/chenhengzh/language_ranker.

## 1   Introduction

Traditional research on enhancing the capabilities of large language models (LLMs) has primarily focused on improving the quality of output distributions through approaches such as scaling up model sizes [1], fine-tuning for specific tasks (SFT) [2, 3], and reinforcement learning with human feedback (RLHF) [4, 5]. However, the decoding process, which converts the output distributions into final responses, has not received sufficient attention. Current decoding strategies, including top-$k$ sampling [6, 7], self-consistency [8], and contrastive decoding [9], are largely rule-based and task-specific, limiting their ability to fully exploit the potential of LLMs' powerful output distributions.

It is found that if an oracle could select the best response from multiple samples generated by a model, the performance of a 7B model could even surpass that of a 70B model as the sample number increases [10] . This finding highlights the tremendous potential of the decoding process in maximizing model performance. To approximate the oracle, recent studies on inference-time computing [11, 12, 13]

---

*Equal contribution.

†Corresponding authors: Zhouchen Lin (zlin@pku.edu.cn) and Yisen Wang (yisen.wang@pku.edu.cn).

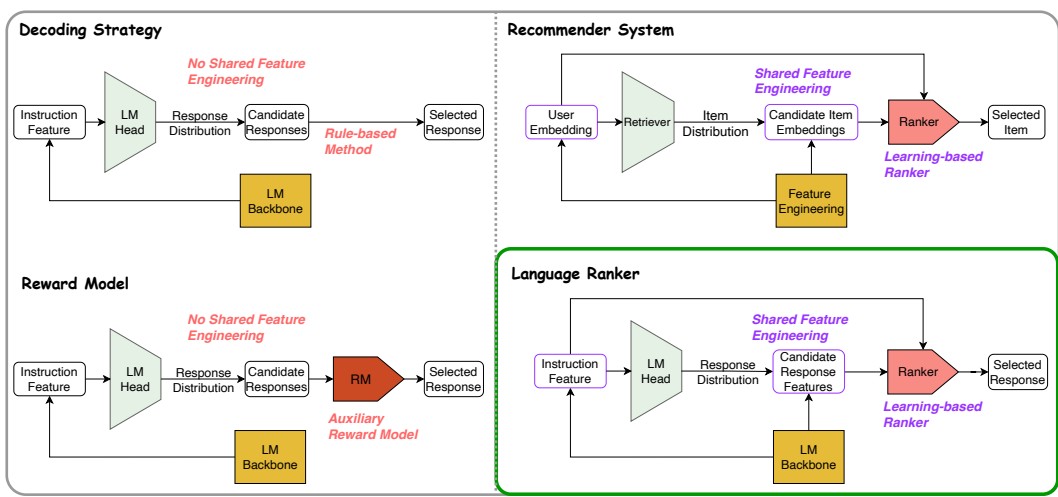

Figure 1: **The comparison among existing methods, recommender system and our Language Ranker.** The two charts on the left highlight the limitations of existing decoding strategies and reward models, in contrast to the recommender system pipeline shown in the top-right. Language Ranker addresses these limitations by incorporating a feature-shared, learnable, and lightweight ranker.

Table 1: Comparison of parameters and performance between the Language Ranker and reward models, using Llama3.1-8B-Instruct as the base model.

| Method | Training Stage | | Inference Stage | | Performance |
|---|---|---|---|---|---|
| | Trainable | GPU-Loaded | Sampling | Ranking | |
| Language Ranker | **<0.5M** | **<0.5M** | 8.2B | **<0.5M** | ★★★ |
| RM (llama8B-LoRA) | 176M | 8.2B | 8.2B | 8.4B | ★★★ |
| RM (gpt2) | 137M | 137M | 8.2B | 137M | ★☆☆ |

have introduced reward models to select the best response. While these methods demonstrate strong performance across various tasks, the reliance on auxiliary reward models significantly increases computational and time overhead during both training and inference, thereby limiting their scalability and applicability in broader contexts.

To address these limitations, we rethink LLMs through the lens of recommender systems. As shown in Figure 1, *each LLM can be viewed as a special recommender system*, where the input serves as the user information, and the model's role is to recommend the most appropriate response as the "item" tailored to the user's needs. Therefore, the model backbone, language head, and decoding process correspond directly to the feature engineering, retriever, and ranker in a traditional recommender system [14]. When a user provides an input, the model backbone first extracts user features, represented as the hidden states of the last token. The language head then generates a coarse response distribution. Finally, a predefined decoding strategy samples several candidate responses from this distribution and selects the most suitable one among them.

In this analogy, the limitations of both existing decoding strategies and reward models become evident. As illustrated in Figure 1, current decoding strategies are typically simple and rule-based, neglecting the crucial role of reranking model responses. Meanwhile, reward models, though effective as rankers, introduce substantial computational overhead both in training and inference. From the perspective of recommender systems, these methods essentially redo the feature engineering for ranking from scratch, ignoring the features already extracted during the recall stage that could have been shared. This redundancy leads to significant unnecessary computations and inefficiency.

In this paper, we propose **Language Ranker**, inspired by recommender systems, to address aforementioned shortcomings of existing methods. The Language Ranker incorporates a carefully designed lightweight ranker to rerank candidate responses generated by the base model, which constitutes the only trainable component in our framework. As illustrated in Figure 2, the earlier layers of the base model can be viewed as shared feature engineering for both the retriever and ranker, similar to

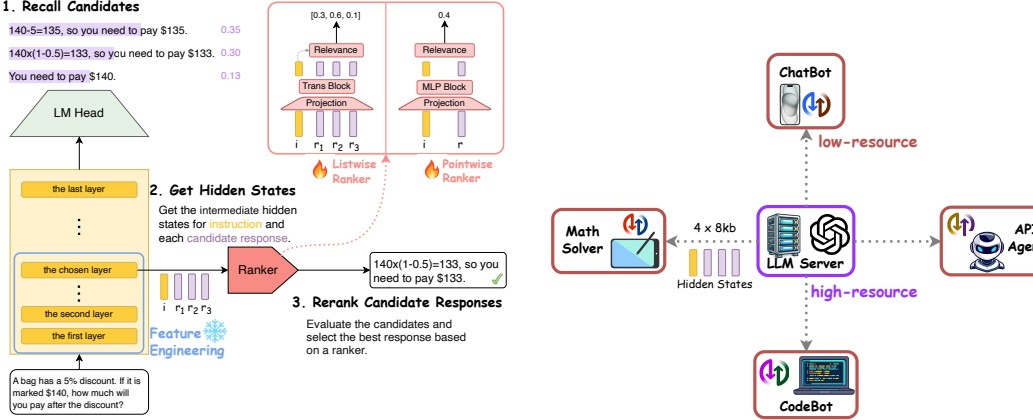

Figure 2: **The Framework of Language Ranker.** The base model first generates multiple candidate responses. The hidden states corresponding to the final tokens of both the instruction and each candidate response are then extracted from a predefined layer and used as features. Finally, the ranker selects the most appropriate response based on these features.

Figure 3: **Personalized Language Ranker.** We can pair a single base model with different rankers to enable personalized adaptation for diverse user needs simultaneously. The base model runs on high-resource central nodes, while rankers can be deployed on edge devices or even local user devices. The CPU-trainability allows each user's ranker to perform continual learning with behavioral data, paving the way for deeper personalization.

recommender systems. Once the candidate responses are sampled, the ranker utilizes the extracted features to rerank the candidates and identify the most appropriate response.

As shown in Table 1, by leveraging the representations of base models, our method achieves performance comparable to that of the Llama8B-based reward model, while requiring only <0.5M additional parameters, significantly reducing the computational overhead during both training and inference stages. Table 3 shows that the lightweight ranker supports both training and inference independently on CPUs, demonstrating the potential to build a personalized Language Ranker. As illustrated in Figure 3, the base model can be paired with different rankers to enhance capabilities across various dimensions. The base model is expected to run on high-resource central nodes, while the rankers can be deployed on edge nodes or even on users' local devices, allowing continual learning and personalized adaptation to diverse user needs.

In conclusion, the main contributions of this paper are:

- We reinterpret LLMs through the lens of recommender systems, revealing the limitations of existing decoding strategies and reward models while highlighting the potential of the decoding process.

- We propose Language Ranker, a novel and lightweight ranking framework for LLMs that is both efficient and effective. It allows a single base model to be flexibly paired with different rankers, allowing personalized adaptation to diverse user needs.

- We conduct extensive experiments across diverse tasks and multiple base models, demonstrating that our framework achieves performance comparable to large-scale reward models while requiring <0.5M additional parameters, thereby substantially reducing computational overhead during both training and inference.

## 2 Decoding as Ranking Mechanisms Design

In this section, we introduce Language Ranker, our lightweight ranking framework for LLMs inspired by recommender system. This framework addresses the limitations of existing decoding strategies and reward models by incorporating an efficient and effective ranking mechanism.

## 2.1 Language Ranker

As shown in Figure 2, we employ a lightweight yet effective ranker to rerank candidate responses generated by language models. Specifically, a hyperparameter is defined to select a specific layer in the model, and the hidden states of this layer are used as features for the ranker. [3] Before inference, the hidden states of the selected layer corresponding to the final token of the given instruction is recorded as the instruction feature, denoted as $i$. The model then begins the inference process, sampling $K$ candidate responses. Once each candidate response is fully generated, the hidden state of the chosen layer corresponding to the final token is recorded as its feature. These features, representing the candidate responses, are denoted as $\{r_k\}_{k=1}^{K}$. These instruction and response features are then fed into the ranker to identify the most suitable response.

Following common practices in recommender systems [14], we design both a listwise ranker and a pointwise ranker to refine candidate responses. Both rankers first project the input features into a low-dimensional space, compressing information while significantly reducing the parameter count for subsequent processing. They then process the projected features using their respective blocks and compute the relevance between each response and the instruction features, which is subsequently used to rerank the responses.

Specifically, the listwise ranker processes all candidates simultaneously, enabling direct comparisons between them:

$$\left[\tilde{i}, \tilde{r_1}, \tilde{r_2}, \cdots, \tilde{r_K}\right] = Trans\left(Proj\left([i, r_1, \cdots, r_K]\right)\right),\tag{1}$$

$$[s_1, s_2, \cdots, s_K] = Rele\left(\tilde{i}, [\tilde{r_1}, \tilde{r_2}, \cdots, \tilde{r_K}]\right).\tag{2}$$

As illustrated in Figure 2, after projection, the instruction feature $i$ and the response features $\{r_k\}_{k=1}^{K}$ interact within a Transformer block. Subsequently, relevance scores between the instruction and each candidate response are computed, and the candidate with the highest score is selected as the final output.

The pointwise ranker, in contrast, evaluates each candidate response individually based on the given instruction feature:

$$\left[\tilde{i}, \tilde{r_k}\right] = \left[MLP\left(Proj(i)\right), MLP\left(Proj(r_k)\right)\right],\tag{3}$$

$$s_k = Rele\left(\tilde{i}, \tilde{r_k}\right).\tag{4}$$

Each projected feature is independently processed using a shared MLP block. The ranker then computes a relevance score between the instruction feature and each response feature. The response with the highest relevance score is selected as the final result.

Notably, the choice of relevance function for both the listwise and pointwise rankers depends on the form of the response labels. If the labels are binary (0 or 1), cosine similarity is applied, framing the task as a classification problem. In contrast, if each response is assigned a specific score, a learnable relevance function is employed to fit these scores:

$$s_k = Rele\left(\tilde{i}, \tilde{r_k}\right),\tag{5}$$

$$= W * concat\left(\tilde{i}, \tilde{r_k}\right).\tag{6}$$

The Transformer block in the listwise ranker and the MLP block in the pointwise ranker can both be extended to multiple blocks. For efficiency, we use a single block in all main experiments, and the impact of the block number is analyzed in Subsection 3.2.

## 2.2 Dataset Construction and Ranker Training

The training dataset for our ranker is constructed in a manner similar to that of reward model datasets [15], introducing virtually no additional computational or time overhead. For each task, we allow the base model to perform sampling and generate 100 responses for each instruction in the training set. During this process, the corresponding instruction features and response features are recorded. These features are then used to train the ranker effectively. After collecting all responses, we assign labels

---

[3]The final layer of the model backbone is often suboptimal for feature extraction; instead, layers located around 60% from the bottom of the model typically yield better representations, as shown in Subsection 3.2.

depending on the characteristics of the task. These details will be discussed within the context of specific tasks in Section 3.

The listwise ranker processes a list of candidates simultaneously, allowing for direct comparisons among them. To prepare the training data, $K$ candidate responses are randomly sampled for each query from the previously constructed dataset. This process is repeated multiple times, and groups that do not contain both positive and negative responses are filtered out. Ultimately, $N$ data groups per query, along with their corresponding hidden states, are collected for training, formally represented as $\left[ i, (r_1^{(n)}, y_1^{(n)}), \cdots, (r_K^{(n)}, y_K^{(n)}) \right]_{n=1}^{N}$. For training process, the loss function is selected based on the form of the labels. If $y_k^{(n)} \in \{0, 1\}$, the task is framed as a classification problem, where $s_k^{(n)}$ is computed using cosine similarity. We optimize the ranker using the following KL divergence loss:

$$\pi_y^{(n)} = \frac{y_k^{(n)}}{\sum_{k=1}^{K} y_k^{(n)}}, \quad \pi_s^{(n)} = \frac{\exp(s_k^{(n)})}{\sum_{k=1}^{K} \exp(s_k^{(n)})}, \tag{7}$$

$$\mathcal{J}_{cls}^{list} = \frac{1}{N} \sum_{n=1}^{N} \mathbb{D}_{KL} \left( \pi_y^{(n)} \| \pi_s^{(n)} \right). \tag{8}$$

If $y_k^{(n)} \in \mathbb{R}$, the task is treated as a regression problem, where $s_k^{(n)}$ is computed by the learnable relevance function previously introduced. We apply the mean squared error (MSE) loss:

$$\mathcal{J}_{reg}^{list} = \frac{1}{N} \sum_{n=1}^{N} \frac{1}{K} \sum_{k=1}^{K} \left( s_k^{(n)} - y_k^{(n)} \right)^2. \tag{9}$$

The pointwise ranker is much simpler than the listwise ranker. For each query, it independently pairs each candidate response with its corresponding instruction, formally represented as: $\left[ i, (r^{(n)}, y^{(n)}) \right]_{n=1}^{N}$. The choice of loss function also depends on the form of the labels. Following the discussion for the listwise ranker, we summarize the corresponding loss functions for the two forms of labels below:

$$p_k^{(n)} = \frac{\exp(s^{(n)})}{1 + \exp(s^{(n)})}, \tag{10}$$

$$\mathcal{J}_{cls}^{point} = -\frac{1}{N} \sum_{n=1}^{N} y^{(n)} \log p^{(n)} + (1 - y^{(n)}) \log(1 - p^{(n)}). \tag{11}$$

$$\mathcal{J}_{reg}^{point} = \frac{1}{N} \sum_{n=1}^{N} \left( s^{(n)} - y^{(n)} \right)^2. \tag{12}$$

## 3 Experiments

In this section, we conduct experiments on three representative LLM tasks: mathematics, coding, and function calling. We further perform detailed analyses and ablation studies, as well as evaluate the transferability of our method. To additionally demonstrate the generality of our approach, we assess its general instruction-following capability, as described in Appendix B.1. The full set of hyperparameters is provided in Appendix A.

### 3.1 Main Experiments

**Baselines** For each task, we train two reward models of different scales for comparison. The first is based on GPT-2 [16], a relatively small model that nevertheless has over 100 times more parameters than our ranker. The second reward model is trained from the corresponding base model using LoRA. Although the number of trainable parameters is similar to GPT-2, the full model must be loaded into GPU memory for both training and inference, resulting in substantially greater computational overhead. In addition, we use the first sampled response from the base model as a simple baseline

Table 2: The total performance across the three tasks compares our methods with reward models and common decoding strategies. The RM means reward model. In the Parameter column, we report the number of trainable parameters for each method. For reward models trained with LoRA, we additionally report the number of GPU-loaded parameters.

| Method | Parameter | MATH | MBPP | xLAM |
|---|---|---|---|---|
| Llama3.1-8B-Instruct | | | | |
| ListRanker (ours) | **0.30M** | **46.3** | 54.5 | 32.6 |
| PointRanker (ours) | **0.28M** | 45.8 | **55.1** | 30.4 |
| RM (gpt2) | 137M | 42.9 | 47.7 | 29.4 |
| RM (Llama8B) | 176M / 8.2B | 45.1 | 52.9 | **32.8** |
| Beam Search | — | 40.3 | 42.3 | 27.0 |
| First Sample | — | 25.1 | 41.9 | 10.6 |
| Qwen2.5-7B-Instruct | | | | |
| ListRanker (ours) | **0.27M** | 74.8 | **63.2** | **71.0** |
| PointRanker (ours) | **0.25M** | **75.2** | 62.7 | 70.4 |
| RM (gpt2) | 137M | 71.9 | 60.2 | 65.4 |
| RM (Qwen7B) | 161M / 7.6B | 74.6 | 62.9 | 70.2 |
| Beam Search | — | 67.9 | 62.2 | 68.0 |
| First Sample | — | 68.7 | 60.6 | 57.0 |
| Qwen2.5-32B-Instruct | | | | |
| ListRanker (ours) | **0.36M** | 81.1 | 74.2 | 72.8 |
| PointRanker (ours) | **0.34M** | **81.3** | 74.6 | 72.4 |
| RM (gpt2) | 137M | 78.8 | 70.6 | 68.8 |
| RM (Qwen32B) | 537M / 32.8B | 80.7 | **75.9** | **73.6** |
| Beam Search | — | 78.1 | 71.4 | 70.6 |
| First Sample | — | 75.9 | 68.2 | 65.2 |

and adopt deterministic beam search as a representative decoding strategy. A detailed analysis and comparison with other decoding strategies are provided in Appendix B.3.

**Ranker Settings**     In all experiments, the rankers are implemented using either a single Transformer block or a single MLP block, and they operate on features extracted from approximately the bottom 60% of the base model's layers. During both training and evaluation, each data group consists of 10 candidate responses. The ranker is trained to classify each response as correct or incorrect, formulating the task as a binary classification problem. Cosine similarity is used to compute the final logits, and the training objective is defined by the classification loss $\mathcal{J}_{cls}$, as specified in Equations 8 or 11.

**Models**     We evaluate our method on LLaMA3.1-8B-Instruct [17], Qwen2.5-7B-Instruct, and Qwen2.5-32B-Instruct [18] to demonstrate its generality across different model architectures and scales. To further validate our approach on a broader range of models, we also conduct experiments on Gemma3-4B-it [19], as presented in Appendix B.2. To ensure fairness, all models are evaluated under the zero-shot setting, with prompts detailed in Appendix D.

**Datasets**     For the mathematics task, we use the MATH dataset [20], which contains 12,500 competition-level problems spanning seven topics and five difficulty levels. To ensure coverage while maintaining efficiency, we uniformly sample 1,000 problems each for training and testing across different topics and difficulty levels. For the coding task, we use the complete MBPP dataset [21], which consists of short Python programming problems, 374 for training and 500 for testing, each paired with test cases to evaluate the correctness of the generated solutions. For the function calling task, we adopt the xlam-function-calling-60k dataset [22], which comprises 60,000 high-quality function calling problems and answers. We randomly sample 1,500 more challenging problems with more than three APIs, and split them into 1,000 training and 500 testing examples.

**Metrics**     For the mathematics, coding, and function calling tasks, we design task-specific labeling criteria to ensure consistent and fair evaluation. For the mathematics task, we extract the final answer

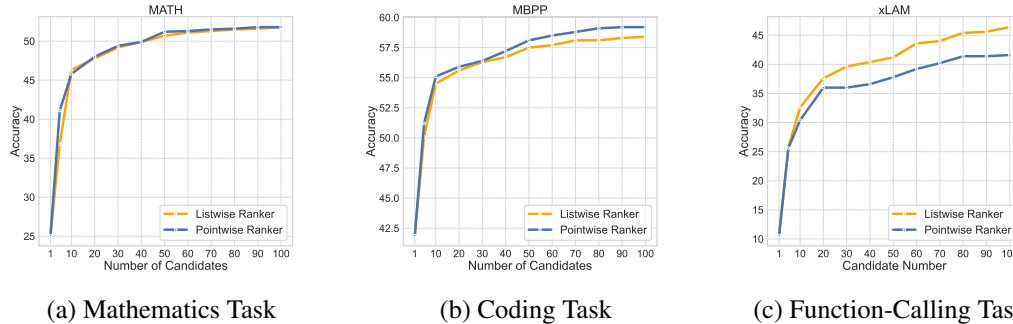

| (a) Mathematics Task | (b) Coding Task | (c) Function-Calling Task |

Figure 4: The performance of the Language Ranker built on Llama3.1 improves consistently across all three tasks as the number of candidate responses increases.

Table 3: The total training time on the MBPP dataset for both CPU and GPU settings, including data loading stages.

| Method | CPU | A100 |
|---|---|---|
| Listwise Ranker | 67s | 44s |
| Pointwise Ranker | 71s | 42s |
| RM (gpt2) | >1h | 72s |
| RM (Llama8b) | too long | 24min |

Table 4: Comparison of performance and parameter on MATH under different ranker architecture ablations, using Llama3.1-8B as the base model.

| Ranker Setting | Accuracy | Parameter |
|---|---|---|
| Listwise Ranker | 46.3 | 0.30M |
| – remove projection | 46.4 | 192M |
| – remove instruction | 44.2 | 0.30M |
| Pointwise Ranker | 45.8 | 0.28M |
| – remove projection | 46.0 | 128M |
| – remove instruction | 44.1 | 0.28M |
| – remove MLP block | 42.5 | 0.25M |

from each response and compare it with the ground-truth answer. A response is labeled as correct if the extracted answer matches exactly; responses that are incorrect or from which the answer cannot be extracted (e.g., due to formatting errors) are labeled as incorrect. For the coding task, we extract the generated code segment and execute it on a set of predefined test cases. A response is labeled as correct only if all test cases pass; otherwise, it is labeled as incorrect. For the function calling task, we extract the function calls from each response and use regular expressions to parse the function names and parameter values. The response is labeled as correct only if both the function and all arguments match the ground-truth function calls.

**Results** Both the listwise and pointwise rankers significantly improve model performance across all tasks. Our lightweight method consistently outperforms the reward model (gpt2), despite being over 100 times smaller in scale, and even achieves performance comparable to reward models trained from the base model. Specifically, for Llama3.1-8B-Instruct, our approach improves over the first-sample baseline by more than 20% on MATH and 12% on MBPP, substantially outperforming both larger reward models. On the function calling task, it trails the Llama8B-based reward model by only 0.2%. For Qwen2.5-7B-Instruct, the Language Ranker outperforms all baselines. For Qwen2.5-32B-Instruct, rankers with fewer than 0.5M parameters achieve performance comparable to 32B-scale reward models, demonstrating remarkable potential. This suggests that rankers can adapt to even larger base models, as the extracted features they rely on become increasingly expressive—allowing them to *stand on the shoulders of giants*.

## 3.2 Analysis and Ablation Study

**Ranker Scaling Law** Figure 4 illustrates the relationship between the performance of the Language Ranker and the number of candidate responses provided to the ranker. We found that performance improves across diverse tasks as the number of candidates increases, demonstrating the Ranker Scaling Law. A key open problem in current research is how to effectively scale inference-time computation in large language models to enhance their performance. Most existing work has focused on optimizing inference configurations during the sampling stage, often relying on traditional reward models for

Table 5: Performance comparison across different ranker configurations in MATH for Llama3.1-8B-Instruct.

| Ranker Type | Hidden States Layer | | | | Block Number | | | |
|---|---|---|---|---|---|---|---|---|
| | 0.1 | 0.3 | 0.6 | 1.0 | 1 | 2 | 3 | 4 |
| Llama3.1-8B-Instruct | | | | | | | | |
| Listwise Ranker | 41.2 | 44.6 | **46.3** | 44.9 | 46.3 | 46.7 | 46.6 | **46.9** |
| Pointwise Ranker | 40.6 | 43.6 | **45.8** | 44.0 | 45.8 | 46.2 | **46.4** | 46.3 |
| Qwen2.5-7B-Instruct | | | | | | | | |
| Listwise Ranker | 70.6 | 72.7 | **74.8** | 73.6 | 74.8 | 74.9 | 75.2 | **75.4** |
| Pointwise Ranker | 71.4 | 73.1 | **75.2** | 73.9 | 75.2 | 75.1 | **75.6** | 75.5 |

Table 6: Performance across different hyperparameter configurations for Llama3.1-8B-Instruct on the MATH dataset. Green indicates the best resuls, while red indicates the worst results.

| Optimizer | SGD | | | | AdamW | |
|---|---|---|---|---|---|---|
| Learning Rate | 0.05 | 0.1 | 0.5 | 1.0 | 1e-5 | 1e-4 |
| Batch Size=256 | 46.2 | 46.1 | 45.8 | 45.7 | 46.2 | 46.1 |
| Batch Size=1024 | 46.1 | 45.9 | 46.3 | 45.9 | 45.9 | 46.1 |

(a) Listwise Ranker

| Optimizer | AdamW | | | |
|---|---|---|---|---|
| Learning Rate | 5e-5 | 1e-4 | 2e-4 | 5e-4 |
| Batch Size=64 | 41.2 | 42.2 | 45.1 | 43.6 |
| Batch Size=256 | 43.2 | 41.9 | 42.8 | 44.7 |

(b) Reward Model (Llama8B)

response reranking [12, 11, 23]. In contrast, our approach focuses on the ranking stage, providing an efficient, effective, and scalable alternative. This distinction underscores the complementarity between our method and sampling-based techniques, suggesting that they can be integrated to further improve model performance.

**CPU Trainability** As shown in Table 3, the lightweight rankers can be efficiently run on CPUs, demonstrating the potential of constructing a personalized Language Ranker. As illustrated in Table 3, the base model can be paired with different rankers to enhance capabilities across various dimensions. The hidden states of the final token ( 8KB) are compact enough to be transmitted over the internet. In this setup, the base model runs on high-resource central nodes, while rankers can be deployed on edge devices or even local user devices, enabling flexible adaptation to diverse user needs. Moreover, the CPU-trainability allows each user's ranker to perform continual learning with behavioral data, paving the way for deeper personalization.

**Ablation Study** To better understand the design of our ranker, we conduct ablation studies on all key components of both the listwise and pointwise architectures. As shown in Table 4, the projection layer compresses high-dimensional features into a lower-dimensional space, playing a critical role in keeping the ranker lightweight. Removing this layer results in a much larger ranker with minimal performance gain. Additionally, we examine the role of the instruction feature, which is used to compute relevance scores with each candidate for ranking. Replacing this feature with a learnable vector leads to a noticeable drop in performance, underscoring the its importance as a form of user information, consistent with our perspective of recommender system.

**Ranker Configurations** The last layer of the model backbone is often not the best choice for providing features. Since the backbone is trained for next-token prediction, the final layers tend to overfit to this specific task. In contrast, intermediate layers typically provide more comprehensive representations of the preceding context, making them better suited for capturing the overall features required for ranking [24]. As shown in the Hidden States Layer part of Table 5, the most effective features for the rankers are extracted from the 60% from the bottom of the model layers. As shown in the Block Number part of Table 5, increasing the ranker's scale has only a marginal impact. Since the base model has already extracted high-quality features, the ranker's task remains relatively simple, making further scaling unnecessary.

**Hyperparameter Robustness** Table 6 presents the hyperparameter robustness of our method, particularly in comparison with reward models. We uniformly sample a series of hyperparameter configurations for both approaches. For the listwise ranker, the accuracy range across 12 configurations is only 0.6%, with the best and worst performances at 46.3% and 45.7%, respectively. In contrast, the

Table 7: The Transfer Performance of a Ranker Trained on a Single Task. All results are tested with Llama3.1-8B-Instruct on the MATH dataset. For task types, we use abbreviations in the table due to space constraints: Prealgebra (PA), Algebra (A), Number Theory (NT), Counting and Probability (CP), Geometry (G), Intermediate Algebra (IA), Precalculus (PC).

| Source Task | Target Task | | | | | | |
|---|---|---|---|---|---|---|---|
| | PA | A | NT | CP | G | IA | PC |
| PA | **67.5** | 61.3 | 38.2 | 43.7 | 33.4 | 21.9 | 32.2 |
| | **0.0** | -0.4 | -0.5 | -0.2 | 0.0 | -0.8 | -1.7 |
| A | 66.0 | **61.7** | 38.5 | 42.0 | 34.7 | 22.3 | 31.1 |
| | -1.5 | **0.0** | -0.2 | -1.9 | -4.0 | -0.4 | -2.8 |
| NT | 64.9 | 60.2 | **38.7** | 41.4 | 35.7 | 20.7 | 31.0 |
| | -2.6 | -1.5 | **0.0** | -2.5 | -2.7 | -2.0 | -2.9 |
| CP | 66.5 | 61.3 | 37.4 | **43.9** | 35.3 | 22.2 | 32.6 |
| | -1.0 | -0.4 | -1.3 | **0.0** | -2.1 | -0.5 | -1.3 |
| G | 66.0 | 60.5 | 36.7 | 41.4 | **37.4** | 22.4 | 31.1 |
| | -1.5 | -1.2 | -1.8 | -2.5 | **0.0** | -0.3 | -2.8 |
| IA | 64.3 | 58.8 | 35.7 | 38.6 | 32.4 | **22.7** | 31.3 |
| | -3.2 | -2.9 | -2.8 | -5.3 | -5.0 | **0.0** | -2.6 |
| PC | 63.0 | 59.1 | 35.6 | 41.1 | 34.5 | 22.3 | **33.9** |
| | -4.5 | -2.6 | -2.9 | -2.9 | -2.9 | -0.4 | **0.0** |

reward model is much more sensitive to hyperparameters, showing a larger variation of 3.9% across 8 configurations (best: 45.1%, worst: 41.2%).

## 3.3 Cross-Domain and Cross-Task Transfer

To assess the generalization capability of the Language Ranker, we conduct transfer experiments on the MATH dataset, which contains seven distinct problem types. We train the ranker on a single problem type and evaluate its generalization to the remaining domains (see Table 7). The results show that rankers trained on any individual domain maintain robust performance across all others. Remarkably, in some cases, the transfer performance approaches that of domain-specific rankers, demonstrating the system's adaptability to unseen domains.

To further evaluate its transferability across broader task domains, we perform cross-task transfer experiments, following a similar experimental setup. Specifically, we train the ranker on a single task type (math or code) and evaluate its generalization ability on the other task. Table 8 shows that rankers trained on any individual task maintain robust performance on the other one. Notably, the transfer performance even surpasses in-domain performance of GPT-2-based reward model (43.4 vs. 42.9 on MATH and 51.2 vs. 47.7 on MBPP), further highlighting our adaptability to unseen domains.

Moreover, unlike recent general reward models, our ranker is highly efficient and easy to train. Its lightweight design allows a single base model to be paired with multiple task-specific rankers, enabling flexible deployment with minimal overhead. This makes the proposed **Language Ranker** framework a practical and scalable solution for adapting to diverse downstream tasks under real-world resource constraints.

## 4 Related Work

**Decoding Methods** A variety of rule-based decoding methods have been proposed to improve language model performance, including top-$k$ sampling [6, 7], temperature-based sampling [25], and nucleus sampling [26]. Beyond these, more refined algorithms have been developed for specific tasks. Self-consistency has been introduced as a method to improve Chain-of-Thought (CoT) reasoning by

Table 8: Cross-task generalization between math and code tasks. Each ranker is trained on one task and evaluated on both to assess transferability.

| Method | To MATH | To MBPP |
|---|---|---|
| Ranker From MATH | **46.3** | 51.2 (-3.3) |
| Ranker From MBPP | 43.4 (-2.9) | **54.5** |
| RM (gpt2) | 42.9 (-3.4) | 47.7 (-6.8) |

majority voting [8, 27]. Other approaches leverage an auxiliary model either selected or fine-tuned to assist in generating responses that better align with desired requirements [28, 9, 29]. However, these methods are typically rule-based or task-specific, which limits both their performance ceiling and application scope. We propose a more general ranking framework to address these limitations.

**Reward Models** Reward models have been widely adopted for the enhancements of LLMs. They serve as learned proxies for human preferences in RLHF [30, 31], and have also been applied to guide multi-step reasoning processes [32, 33]. While effective in a range of scenarios, reward models typically introduce significant computational overhead, limiting their practical deployment in real-world systems. To address this issue, some efforts aim to teach models to act as self-critics [34, 35], but their performance remains suboptimal. An embedding-based alternative has been proposed to simplify reward model training [36], but it primarily targets RLHF settings and still requires an additional forward pass during both training and inference to extract embeddings. Another line of work proposes scoring all candidate tokens simultaneously to reduce call frequency, but relying more heavily on large-scale models [37].

**Inference-Time Computing** Recently, there has been growing interest in scaling inference-time computation to improve the performance of LLMs. Most existing approaches focus on optimizing configurations at the sampling stage, often relying on traditional reward models to evaluate and rerank generated responses [13, 38, 23, 15, 39], or specific design on extending CoT length [40, 41, 42]. In contrast, our method shifts the focus to the ranking stage and introduces a lightweight architecture that operates directly on features already extracted by the base model. This design eliminates the need for additional forward passes, providing a scalable, efficient, and effective alternative. We believe our approach complements sampling-based strategies and can be combined with them to further improve model performance.

## 5 Conclusion and Discussion

In this paper, we have introduced the Language Ranker, a novel lightweight ranking framework for enhancing the LLMs. By rethinking LLMs through the lens of recommender systems, we identified the limitations of existing decoding strategies and reward models, particularly their limitations on rule-based methods and computationally expensive reranking. Our approach integrates a lightweight ranker that leverages features extracted by the base model, enabling more efficient, effective and scalable reranking with minimal computational overhead.

Moreover, the separability of the ranker from the base model further enhances the flexibility of our approach. This decoupling allows for the independent optimization of the ranker, enabling a base model to be paired with multiple rankers that enhance different aspects of its capabilities, making it adaptable to various domains. We hope that our approach can offer new perspectives for the future of language model inference and contribute to the development of more resource-efficient AI systems.

## Acknowledge

Zhouchen Lin is supported by the NSF China (No. 62276004), the Beijing Major Science and Technology Project under Contract no. Z251100008425006 and the State Key Laboratory of General Artificial Intelligence. Yisen Wang is supported by Beijing Natural Science Foundation (L257007), Beijing Major Science and Technology Project under Contract no. Z251100008425006, National Natural Science Foundation of China (92370129, 62376010), Beijing Nova Program (20230484344, 20240484642), and State Key Laboratory of General Artificial Intelligence.

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

# A    Hyperparameter settings

In sampling process, we set temperature as 1.5 for diverse responses and max_new_tokens as 1024 to make sure completed answers. We sample 100 responses for each problem. During training, we perform a grid search over the parameter ranges specified in Table 9.

Table 9: The hyperparameter list

| Hyperparameter | Value |
|---|---|
| *Sampling* | |
| Sampling Temperature | 1.5 |
| Sampling Max New Tokens | 1024 |
| *Ranker Training* | |
| Batch Size | [256, 1024] |
| Epoch | 1 |
| Optimizer | [SGD, AdamW] |
| SGD LR | [0.05, 0.1, 0.5, 1.0] |
| SGD Momentum | [0.0, 0.9] |
| AdamW LR | [1e-5, 1e-4] |
| AdamW Betas | (0.9, 0.999) |
| Weight Decay | 1e-4 |
| LR Schedule | [Constant, Cosine Decay] |
| Projection Dimension | 64 |
| *Reward Model Training* | |
| Batch Size | [64, 256] |
| Epoch | 1 |
| Optimizer | AdamW |
| AdamW LR | [5e-5, 5e-4] |
| AdamW Betas | (0.9, 0.999) |
| Weight Decay | 1e-4 |
| LR Schedule | [Constant, Cosine, Decay] |
| LoRA r | 64 |
| LoRA alpha | [64, 128] |

# B    Additional Experimental Results

## B.1    Instruction-Following Task

Our framework performs well on the three tasks presented in Section 3. To further demonstrate its general applicability, we also evaluate it on a mixed instruction-following task.

**Models**    Considering that instruct models are specifically fine-tuned for instruction-following tasks, we conduct evaluations on this task with Llama3.1-8B-Base [17] and Qwen2.5-7B-Base [18]. For fairness, all models are evaluated using zero-shot prompts, as shown in Appendix D.

**Datasets**    We use the first 1,000 queries from the Databricks-Dolly-15k dataset [43] for training. For evaluation, we adopt AlpacaEval [44], a widely recognized benchmark for assessing instruction-following capabilities in LLMs. It consists of diverse test queries sourced from Self-Instruct, OASST, Anthropic's Helpful dataset, Vicuna, and Koala.

**Metrics**    Unlike tasks such as mathematics, instruction-following lacks objective ground-truth answers, making rule-based evaluation infeasible. To address this, we follow the AlpacaFarm [44] method and prompt DeepSeek-V3 to simulate human judgment by assigning scores from 0 to 5 to all sampled responses. These responses are then used to train both our ranker and reward models. For evaluation, we adopt the official AlpacaEval evaluator to compute the Length-Controlled Win Rate metric [45], a relative measure based on a reference model. For each base model, we use its corresponding instruct variant as the reference—for example, Llama3.1-8B-Instruct for Llama3.1-8B-Base.

Table 10: The evaluation on general instruction-following tasks compares our method against reward models and common decoding strategies. We conduct experiments on Llama3.1-8B-Base and Qwen2.5-7B-Base, reporting the win rate using the corresponding Instruct model as the reference. RM (base) refers to reward models trained from the respective base model.

| Method | Parameter | Llama3.1-8B-Base | Qwen2.5-7B-Base |
|---|---|---|---|
| ListRanker (ours) | <0.3M | 30.7 | **46.3** |
| PointRanker (ours) | <0.3M | 27.1 | 45.8 |
| RM (gpt2) | 137M | 27.1 | 42.9 |
| RM (base) | ~170M/8B | **31.6** | 45.3 |
| First Sample | — | 19.0 | 25.1 |
| Beam Search | — | 20.4 | 40.3 |

Table 11: Comparison between our methods and reward models on math and code tasks. The RM denotes reward model. The Parameter column reports the number of trainable parameters for each method. For reward models trained with LoRA, we also report the number of GPU-loaded parameters.

| Method | Parameter | MATH | MBPP |
|---|---|---|---|
| Gemma3-4B-it | | | |
| ListRanker (ours) | **0.20M** | 72.2 | 51.4 |
| PointRanker (ours) | **0.19M** | **72.4** | **51.7** |
| RM (gpt2) | 137M | 69.2 | 50.3 |
| RM (Gemma3) | 161M / 7.6B | 69.1 | 50.9 |
| Beam Search | — | 67.4 | 49.2 |
| First Sample | — | 63.5 | 48.8 |

**Results** As shown in Table 2, the performance of our methods far exceeds that of vanilla decoding strategies and is comparable to the reward models trained from base models. Notably, with the assistance of a 0.3M-level ranker, Qwen2.5-7B-Base achieves a 46.3% win rate compared to Qwen2.5-7B-Instruct, which has undergone extensive fine-tuning on various instruction-following tasks.

## B.2 Experimental Results on Gemma3-4B-it

We conduct experiments across a wide range of tasks using both 7B and 32B base models, covering two prevalent architectures: LLaMA and Qwen. To further validate the generality of our approach, we additionally include results on Gemma3-4B-it [19], thereby expanding the evaluation across both model scales and backbone architectures. The results for the math and code tasks are presented in Table 11.

Consistent with the findings in Section 3, both the listwise and pointwise rankers substantially improve performance on the math and code tasks with Gemma3-4B-it. These experiments further confirm the effectiveness and robustness of our method across different model sizes and architectures.

## B.3 Detailed Comparison with Existing Decoding Methods

We clarify that decoding methods generally fall into two broad categories: (i) reranking-based methods, which operate on sampled responses as our method does, and (ii) methods that modify the model's output probability distribution during generation.

The first category often relies on auxiliary reward models like our baselines, or is task-specific. For example, self consistency [8] improves reasoning performance by aggregating multiple sampled answers via majority voting. This technique is particularly effective in tasks like math, where the final answer is often a short, well-defined value—making consensus across candidates meaningful.

However, for tasks such as code generation, function calling, or even general instruction-following, the outputs tend to be long, diverse, and semantically equivalent in multiple ways. In these settings,

Table 12: Comparison between our ListRanker and self consistency methods across different tasks. Self consistency improves reasoning tasks like math but performs weakly on code, function calling, and instruction-following tasks.

| Method | MATH | MBPP | xLAM | AlpacaEval |
|---|---|---|---|---|
| ListRanker (ours) | **46.3** | **54.5** | **32.6** | **30.7** |
| Self-Consistency | 44.9 | 41.9 | 24.6 | 20.4 |
| First Sample | 25.1 | 41.9 | 10.6 | 20.4 |

candidate responses often differ significantly in surface form, making majority voting unreliable and diminishing the effectiveness of self consistency. As shown in Table 12, self consistency indeed improves performance on MATH, but shows much weaker performance on function calling, and negligible gains on code and instruction-following tasks.

The second category includes methods such as Contrastive Decoding [9] and DoLa [46], which refine the model's output distribution during generation. ***These approaches are orthogonal to ours***, which focuses on post-sampling ranking. They are complementary to our method and baselines, and can be integrated independently; therefore, we do not include them as direct comparisons in our evaluation.

## C   Limitations

The proposed ranker demonstrates strong effectiveness, efficiency, and transferability. However, it operates on the hidden states of the base model, requiring access to the final-token representations during inference. This requirement introduces no computational overhead theoretically, and storing the representation of only the final token incurs minimal memory usage. Nevertheless, it is not yet fully supported by widely used inference frameworks such as vLLM. We believe that, with the rapid progress of representation-based methods, this limitation will be gradually alleviated in the near future.

# D   Prompts for Each Task

## Prompt for mathematics task

**system:**
You are a math expert.

**user:**
Please solve the given math problem step by step and present the answer in the following format: "\boxed{X}", where X is the answer.
{Question}

## Prompt for coding task

**system:**
You are an expert Python programmer.

**user:**
Write a Python function based on the following instructions and test example. Please ensure that the function is clearly marked with a start and end so I can easily extract it from your output.

Instructions:
{question}

Test Example:
{test_list[0]}

Please provide your code with clear start and end markers, like so:

```
#START OF CODE
def {function_name(input)}:
    ... function code ...
    return result
#END OF CODE
```

## Prompt for function calling task

**system:**
You are a function-calling assistant. Your role is to complete tasks solely through correct function calls, without generating any additional text. For each task, directly output the function call(s) required to complete it. If the task involves multiple steps, you may issue multiple function calls sequentially. Each function call must be formatted as a JSON object. For example: [{"name": "functionA", "arguments": {"param1": "value1", "param2": "value2"}}, {"name": "functionB", "arguments":{"param1": "value1", "param2": "value2"}}]
The following are the available functions: {function_list}

Now, use the appropriate function(s) to complete the given task.

**user:**
{Question}
Please directly output the function call(s) to solve the task without any other text.

## Prompt for instruction-following task

**system:**
You are an assistant.

**user:**
Below is an instruction that describes a task, paired with an input that provides further context. Write a response that appropriately completes the request.
{Instruction} Begin!

## Scoring criteria in instruction-following task

Review the user's question and the corresponding response using the additive 5-point scoring system described below

The user's question is between <question>and </question>The response of the AI Assistant is between <response>and </response>

Points are accumulated based on the satisfaction of each criterion: - Add 1 point if the response is relevant and provides some information related to the user's inquiry, even if it is incomplete or contains some irrelevant content. - Add another point if the response addresses a substantial portion of the user's question, but does not completely resolve the query or provide a direct answer. - Award a third point if the response answers the basic elements of the user's question in a useful way, regardless of whether it seems to have been written by an AI Assistant or if it has elements typically found in blogs or search results. - Grant a fourth point if the response is clearly written from an AI Assistant's perspective, addressing the user's question directly and comprehensively, and is well-organized and helpful, even if there is slight room for improvement in clarity, conciseness or focus. - Bestow a fifth point for a response that is impeccably tailored to the user's question by an AI Assistant, without extraneous information, reflecting expert knowledge, and demonstrating a high-quality, engaging, and insightful answer. - If the response repeats itself or is not concise and to the point, score the response 0.

<question>prompt</question>
<response>response</response>

After examining the user's instruction and the response: - output the score of the evaluation using this exact format: "score: <total points>", where <total points>is between 0 and 5 - Briefly justify your total score, up to 100 words.

