# OpenReview forum: "Language Ranker: A Lightweight Ranking framework for LLM Decoding"
_NeurIPS.cc/2025/Conference — NeurIPS 2025 poster_

### Official Review · Reviewer_pdDQ · 2025-06-10

[review text omitted: it was posted to a different submission]

---

> ### Author Rebuttal · Authors · 2025-07-31
>
> Dear Reviewer pdDQ,
>
> Thank you for your time and effort in reviewing our paper.
>
> However, we noticed a significant mismatch between your comments and the content of our submission — to the extent that we are concerned the review may have been mistakenly attached from a different paper. Specifically, your review describes our method as a **multi-agent system with agent communication and coordination**, whereas our work proposes a lightweight ranking framework for refining responses from a **single language model, without involving any multi-agent interactions.**
>
> If there are any concerns related to our actual work, we would be more than happy to address them.
>
> Thank you again for your consideration.
>
> Sincerely,
>
> *The Authors*

---

### Official Review · Reviewer_gdjF · 2025-06-27

**Clarity:** 2
**Significance:** 2
**Originality:** 2
**Rating:** 4
**Confidence:** 4

**Summary:**

This paper introduces “Language System”,  a novel and lightweight ranking framework designed to enhance Large Language Models (LLMs) by reinterpreting the decoding process through the lens of recommender systems. The authors argue that existing decoding strategies and auxiliary reward models suffer from high computational costs and redundancy. Language System addresses this by employing a lightweight ranker that leverages features extracted directly from the base model to rerank candidate responses. The framework aims to provide comparable performance to large-scale reward models while significantly reducing additional parameters and computational overhead during training and inference.

**Questions:**

1.Can the authors provide a more rigorous theoretical analysis or a deeper conceptual justification for why the recommender system analogy is particularly effective for LLM decoding, beyond just an empirical observation of efficiency? What specific properties of recommender systems are being leveraged that are uniquely beneficial here?
2.Could the authors provide more specific details on the dataset construction for each task, especially regarding how the labels for candidate responses are generated?
3.The method involves generating K candidate responses. How robust is the performance of Language System to the size of this candidate set (K)? Is there an optimal K, and how does varying K affect both performance and efficiency?

**Ethical Concerns:**

["NO or VERY MINOR ethics concerns only"]

**Final Justification:**

The improvement in methodological rigor is commendable: The authors provided detailed explanations of the dataset construction and response labeling process, which increases my confidence in the rigor of this work. Their analysis of the candidate set size K also shows they considered the method's practical application. These are positive improvements that enhance the overall quality of the paper.

**Limitations:**

Yes.

**Quality:**

2

**Strengths And Weaknesses:**

Strengths:   The paper introduces a compelling analogy between LLM decoding and recommender systems, offering a fresh perspective on the problem of efficient LLM inference.  The proposed Language System is highly parameter-efficient, requiring only <0.5M additional parameters, which significantly reduces computational overhead during both training and inference compared to large reward models. The experimental results demonstrate that Language System achieves performance comparable to large-scale reward models across various tasks.


Weaknesses: While the recommender system analogy is interesting, the core technical contribution of employing a lightweight ranker on extracted intermediate features, while efficient, may not be sufficiently novel. Lack of Deep Theoretical Analysis: The paper primarily presents an analogy and empirical results without delving into a deeper theoretical justification for why this specific recommender system perspective offers unique advantages beyond empirical efficiency.The paper primarily compares against reward models and simple decoding strategies. A more comprehensive comparison against other state-of-the-art efficient decoding or reranking methods that do not rely on auxiliary reward models would be crucial to fully assess the novelty and superiority of the proposed approach.

---

> ### Author Rebuttal · Authors · 2025-07-31
>
> We appreciate your recognition of the **fresh perspective, strong empirical performance, and high efficiency of our proposed framework**. The concerns raised—particularly regarding theoretical justification and evaluation breadth—are valuable and well taken. We address each of these points in detail below with further clarifications, analysis, and new experimental evidence.
>
> ---
>
> **W1.** Lack of novelty.
>
> We respectfully disagree with the concern regarding novelty. As noted by reviewers **RWdC** and **L4gf**, **novelty is one of the key strengths** of our work. Specifically, our method introduces **a series of novel contributions** that clearly distinguish it from existing approaches:
>
> 1. **Purpose of Feature Reuse:** We are the **first to use LLM hidden states explicitly for response reranking** as an **efficient and effective inference-time method**.
> 2. **Alignment Between Instruction and Response Features:** Inspired by the recommender system perspective, we treat the final token of the **instruction** as a **user feature** and compute its relevance to each candidate response’s feature. This is the **first work that applies recommender system principles to LLMs.** As shown in **Table 4**, our ablation study demonstrates that this design significantly improves performance, highlighting the importance of modeling user–response relevance explicitly.
> 3. **Personalized Language System:** Our framework allows a single base model to be paired with **multiple lightweight rankers**, each tailored to different user preferences or task requirements — analogous to personalized ranking in recommender systems [1]**.** This enables a novel form of **personalized LLM system**, which prior approaches do not efficiently achieve, especially under real-world resource constraints.
>
> [1] Pei, C., et al. Personalized Re-ranking for Recommendation.
>
> ---
>
> **W2.** Lack of Deep Theoretical Analysis.
>
> Thank you for this constructive suggestion. We agree that a theoretical perspective is valuable, and we have developed a formal analysis that explains **why the two‑stage Language System, inspired by recommender systems, offers unique advantages beyond empirical results**. Below, we present the core insights of this analysis.
>
> ### **Modelling LLM Response Generation as Ultra-Large Classification**
>
> Let $\mathcal{V}$ be the vocabulary ($|\mathcal{V}|=V$) and $L$ the maximum response length. A language model $h_{\theta}$ induces a distribution:
>
> $h_{\theta}(y\mid x)\;\mathcal{X}\longrightarrow\Delta\bigl(\mathcal{Y}\bigr), \quad\text{with } \mathcal{Y}= \mathcal{V}^{L}, |\mathcal{Y}|=V^{L}.$
>
> Generating a single response can therefore be viewed as a $|\mathcal{Y}|$-way classification task. Suppose the base model currently has an error rate $\varepsilon\in(0,1)$, and our target is to achive a stricter error $\tau<\varepsilon$.
>
> There are two strategies:
>
> 1. **Scheme I:** further fine-tuning $h_{\theta}$;
> 2. **Scheme II (ours):** drawing $K$ independent samples from $h_{\theta}$ and training a $K$-way ranker $g_{\phi}$.
>
> ### **Theorem 1 (Sample-Efficiency of Top-$K$ Reranking)**
>
> Let $d_1=\Theta(L\log V)$ denote the VC dimensions of the hypothesis classes for Scheme I and $d_{2}=\Theta(\log K)$ for Scheme II.
>
> If $\varepsilon^{K}<\tfrac{\tau}{2}\;$ and $d_{2}=o\bigl(d_1\bigr) \tag{A}$,
>
> then for any confidence $\delta\in(0,1)$ the additional labeled samples required satisfy
>
> $m_{\text{II}} = \tilde{\mathcal{O}}\Bigl(\tfrac{d_2+\ln(1/\delta)}{(\tau-\varepsilon^{K})^{2}}\Bigr) = o\Bigl( \tfrac{d_{1}+\ln(1/\delta)}{(\varepsilon-\tau)^{2}} \Bigr) = o\bigl(m_{\text{I}}\bigr).$
>
> That is, Scheme II **requires strictly fewer labeled samples than Scheme I** to reach the same target error.
>
> ### **Proof Sketch**
>
> - **Scheme I:** Standard PAC–VC bounds yield
>
>     $m_{\text{I}} = \tilde{\mathcal{O}}\left(\frac{d_{1} + \ln \frac{1}{\delta}}{(\varepsilon - \tau)^{2}}\right).$
>
> - **Scheme II:** Top-$K$ sampling misses the correct label with probability $\varepsilon^{K}$. If the ranker errs with probability $\varepsilon_{2}$, the overall error is
>
>     $E_{\text{II}} = \varepsilon^{K} + (1 - \varepsilon^{K})\varepsilon_{2}.$
>
>     Imposing $E_{\text{II}} \le \tau$ gives $\varepsilon_{2} \le  \frac{\tau - \varepsilon^{K}}{1 - \varepsilon^{K}} $.
>
>     Applying PAC-VC bounds for the ranker:
>
>     $m_{\text{II}} = \tilde{\mathcal{O}}\left(\frac{d_{2} + \ln \frac{1}{\delta}}{\varepsilon_2^{2}}\right) = \tilde{\mathcal{O}}\left(\frac{d_{2} + \ln \frac{1}{\delta}}{(\tau - \varepsilon^{K})^{2}}\right).$
>
>     Since $d_{2} \ll d_{1}$, this implies $m_{\text{II}} \ll m_{\text{I}}$.
>
>
> ### **Remarks**
>
> - **Typical LLM Regime:** For realistic setting $V \sim 10^{5}$, $L \approx 100$, and moderate $K \le 10$, we have $d_{2} \ll d_{1}$. Pre-trained LLMs often achieve $\varepsilon < 0.4$; with $K = 5$ the coverage condition $\varepsilon^{K} < \tau/2$ is easily met for practical $\tau$ (e.g., 5%).
> - **Theoretical Justification:** Given that $d_{2} << d_{1}$ and $m_{\text{II}} \ll m_{\text{I}}$. **Scheme II** requires significantly fewer training and inference parameters, as well as substantially less training data. This provides a strong theoretical justification for the efficiency and practicality of our proposed method.
>
>
> ---
>
> **W3.**  Incomplete comparison with decoding methods.
>
> Thank you for the suggestion. We clarify that decoding methods generally fall into two broad categories: (i) **reranking-based methods**, which operate on sampled responses as our method does, and (ii) methods that **modify the model’s output probability distribution** during generation.
>
> The first category often relies on **auxiliary reward models** like our baselines, or is **task-specific**. For example, **self consistency** [2] improves reasoning performance by aggregating multiple sampled answers via majority voting. This technique is particularly effective in tasks like **math**, where the final answer is often a short, well-defined value—making consensus across candidates meaningful.
>
> However, for tasks such as **code generation**, **tool use**, or even **general instruction-following** (see Appendix B), the outputs tend to be long, diverse, and semantically equivalent in multiple ways. In these settings, candidate responses often differ significantly in surface form, making majority voting unreliable and diminishing the effectiveness of self consistency.
>
> As shown in the table below, self consistency indeed improves performance on **MATH**, but shows **much weaker performance on tool use**, and **negligible gains on code and instruction-following** tasks.
>
> |  | MATH | MBPP | xLAM | AlpacaEval |
> | --- | --- | --- | --- | --- |
> | ListRanker (ours) | **46.3** | **54.5** | **32.6** | **30.7** |
> | Self-Consistency | 44.9 | 41.9 | 24.6 | 20.4 |
> | First Sample | 25.1 | 41.9 | 10.6 | 20.4 |
>
> The second category includes methods such as Contrastive Decoding [3] and DoLa [4], which refine the model’s output distribution during generation. These approaches are **orthogonal to ours**, which focuses on post-sampling ranking. They are complementary to our method and baselines, and can be integrated independently; therefore, we do not include them as direct comparisons in our evaluation.
>
> [1] Chen,X., et al, RM-R1: Reward Modeling as Reasoning.
>
> [2] Wang, X., el al. Self-Consistency Improves Chain of Thought Reasoning in Language Models.
>
> [3] Li, X., et al. Contrastive Decoding: Open-ended Text Generation as Optimization.
>
> [4] Chuang, Y., et al. DoLa: Decoding by Contrasting Layers Improves Factuality in Large Language Models.
>
> ---
>
> **Q1.** Provide a theoretical analysis.
>
> Thank you for your suggestion. We provide the theoretical analysis in our response to **W2**. In summary, under the setting of enhancing LLM performance on a specific downstream task, we show that our **two-stage Language System** can achieve comparable or better target performance with **significantly lower sample complexity and training cost** compared to directly fine-tuning the LLM.
>
> ---
>
> **Q2.** Provide details on the dataset construction, especially response labeling.
>
> Thank you for the question. The construction of training datasets for each task involves two steps:
>
> 1. **Response Sampling:** We prompt the base language model to generate **100 candidate responses** for each input query.
> 2. **Response Labeling:** Each response is labeled based on task-specific criteria.
>
> Specifically, the labeling criteria for different tasks are as follows:
>
> - **Math:** We extract the final answer from each response and compare it with the ground-truth answer. A response is labeled as correct if the extracted answer matches exactly. Responses that are incorrect or from which the answer cannot be extracted (due to formatting errors) are labeled as incorrect.
> - **Code:** We extract the code segment from each response and execute it on a set of predefined test cases. A response is labeled as correct only if **all test cases pass**; otherwise, it is labeled as incorrect.
> - **Tool Use:** We extract the tool call from each response and use regular expressions to parse the **tool name** and **parameter values**. The response is labeled as correct only if **both the tool and all arguments** match the ground-truth tool calls.
>
> ---
>
> **Q3.** How robust is the performance of Language System to the size of this candidate set (K)?
>
> As shown in **Figure 4**, we evaluate the ranker’s performance across different numbers of candidate responses (ranging from 1 to 100). We observe that performance generally improves as the number of candidates increases, but our method remains effective even with a **small candidate set**.
>
> To balance effectiveness and efficiency, we set K = 10 as the default configuration for our main experiments. This setting achieves strong performance while keeping computational overhead low, making it suitable for practical deployment.

---

> ### Author Response · Authors · 2025-08-05
> **Request for Further Discussion**
>
> Dear Reviewer gdjF,
>
> We apologize for the interruption and would like to kindly request your further discussion and guidance regarding our work.
>
> In our response, we have provided detailed explanations to address your concerns. For example, we included a **theoretical analysis** highlighting the advantages of our recommender‑system‑inspired method, and we added further discussions and results covering **additional decoding methods**.
>
> We sincerely thank you again for your thorough review and constructive feedback, and we very much hope to engage in further discussion with you.
>
> Sincerely,
> The Authors

---

> ### Author Response · Authors · 2025-08-06
>
> Dear Reviewer gdjF,
>
> Thank you for your time and efforts for reviewing our paper.
>
> We noticed that you have submitted the final justification, but it is not visible to us. We would like to kindly confirm whether our response has addressed your concerns. If you have any further concerns, we would be happy to address them.
>
> Sincerely,
> The Authors

---

### Official Review · Reviewer_L4gf · 2025-07-01

**Clarity:** 2
**Significance:** 3
**Originality:** 3
**Rating:** 4
**Confidence:** 4

**Summary:**

This paper introduces Language System, a lightweight ranking framework inspired by recommender systems to improve the decoding phase of large language models. The authors propose to treat the decoding process as a ranking problem, akin to recommender system pipelines, and design a learnable, compact ranker to rerank sampled responses using features from intermediate LLM layers. The method requires <0.5M parameters and eliminates the need for expensive reward model inference. Experiments across math, coding, function-calling, and instruction-following tasks demonstrate that Language System achieves comparable or superior performance to much larger reward model-based approaches.

**Questions:**

Q1: How would the performance be affected if only 10–20 candidate responses were sampled instead of 100? Can the ranker still outperform reward models under realistic latency budgets?

Q2: The paper mentions CPU trainability. Are there any preliminary results or benchmarks on on-device adaptation?

Q3: Would it be feasible to integrate sampling and ranking into a unified training loop, where the ranker could guide sampling directly (e.g., via rejection sampling or contrastive decoding)? Or it is possible that such "ranker" is also beneficial in pretraining data selection?

**Ethical Concerns:**

["NO or VERY MINOR ethics concerns only"]

**Final Justification:**

I believe the authors' rebuttal has addressed most of my concerns. After carefully reviewing the interactions between the authors and other reviewers, I think a score of 4 is fair, so I will maintain my initial score.

**Limitations:**

The method still requires storing and retrieving large intermediate hidden states, which incurs I/O overhead. While the ranker is light, the pipeline depends on sampling a large number of outputs (K=100), which could challenge real-time or low-resource applications.

**Quality:**

3

**Strengths And Weaknesses:**

## Strength.

- **Novel perspective.**  The paper offers a fresh conceptualization of LLM decoding as a recommendation-ranking problem, which is both intuitive and practical.
- **Efficient architecture.** The proposed rankers (listwise and pointwise) are extremely lightweight (<0.5M parameters), enabling fast and cost-effective reranking.
- **Strong empirical results.** Across multiple tasks and model sizes (LLaMA3, Qwen2.5), the method consistently matches or outperforms larger reward models like GPT-2 and LoRA-trained variants.

---

## Weaknesses.

- **W1: Insufficient baselines**

The paper does not compare the proposed method against several state-of-the-art open-source reward models, such as generative reward models. Additionally, inference-time scaling techniques like self-consistency are only briefly mentioned and not empirically evaluated. Without these comparisons, it is difficult to assess where the method stands relative to the current best practices in decoding optimization.

- **W2: Limited model coverage.**

Although the method is validated on LLaMA and Qwen series models, it remains unclear whether it generalizes across different architecture families (e.g., Gemma, Mistral) or performs well on smaller-scale models (<7B). This restricts the generalizability claims, especially for edge or low-resource deployment scenarios.


- **W3: Narrow scope of transferability.**

The transferability experiments are limited to different subdomains within the math task. A more compelling demonstration would involve testing cross-domain transfer—e.g., training on math and evaluating on code—to align with recent trends in general reward models that aim to serve multiple downstream tasks without retraining.

- **W4: Missing ablation on ranker scale.**

The paper does not investigate whether increasing the size or complexity of the ranker (e.g., using larger projection dimensions or more Transformer blocks) leads to consistent performance gains. Or does other architecture also effective? Understanding the scaling behavior of the ranker is essential for practitioners seeking the best trade-off between efficiency and performance.

---

> ### Author Rebuttal · Authors · 2025-07-31
>
> We sincerely appreciate your recognition of the **novelty of our approach**, as well as its **strong empirical results** and **efficient design**. The concerns raised—mainly regarding evaluation scope and generalizability, with some minor misunderstandings—are addressed below through additional experiments and clarifications.
>
> ---
>
> **W1.** **Insufficient baselines.**
>
> Thank you for your constructive suggestion. We address the concerns below with further experiments and analysis.
>
> **Generative Reward Model (GRM)**
>
> We include additional results on **LLaMA3.1–8B-Instruct** to compare our method with strong GRMs. GRM performance remains heavily constrained by their training data. For example, on the **math** task—where **RM-R1** is enhanced via RL—it slightly outperforms our method by **1.5%**. However, on out-of-domain tasks like **tool use**, its performance drops significantly, lagging behind ours by **5.8%**.
>
> Moreover, GRMs are **computationally expensive** to train [1], and continual fine-tuning for new domains is often impractical. This makes them less flexible and scalable than our **lightweight** approach.
>
> |  | Parameter | MATH | xLAM |
> | --- | --- | --- | --- |
> | ListRanker (ours) | **0.3M** | 46.3 (-1.5) | **32.6** (+5.8) |
> | RM-R1-DS-7B | 7.6B | **47.8** | 26.8 |
> | RM-R1-Qwen-7B | 7.6B | 45.2 | 26.4 |
> | First Sample | — | 25.1 | 10.6 |
>
> In terms of ranking-time overhead, GRMs are also nearly impractical. Take the **math** task as an example: our method introduces **negligible latency** during ranking, whereas GRMs require generating over **16K tokens per query** to compute scores—resulting in **over one minute of delay**, which is virtually unacceptable in real-world applications.
>
> |  | Parameter | Ranking Tokens | Ranking Latency |
> | --- | --- | --- | --- |
> | ListRanker (ours) | **0.3M** | — | **< 1s** |
> | RM-R1-DS-7B | 7.6B | 16.1K | > 1m |
> | RM-R1-Qwen-7B | 7.6B | 18.3K | > 1m |
>
> **Decoding Optimization**
>
> We clarify that decoding methods generally fall into two broad categories: (i) **reranking-based methods**, which operate on sampled responses as our method does, and (ii) methods that **modify the model’s output probability distribution** during generation.
>
> The first category often relies on **auxiliary reward models** like our baselines, or is **task-specific**. For example, **self consistency** [2] improves reasoning performance by aggregating multiple sampled answers via majority voting. This technique is particularly effective in tasks like **math**, where the final answer is often a short, well-defined value—making consensus across candidates meaningful.
>
> However, for tasks such as **code generation**, **tool use**, or even **general instruction-following** (see Appendix B), the outputs tend to be long, diverse, and semantically equivalent in multiple ways. In these settings, candidate responses often differ significantly in surface form, making majority voting unreliable and diminishing the effectiveness of self consistency.
>
> As shown in the table below, self consistency indeed improves performance on **MATH**, but shows **much weaker performance on tool use**, and **negligible gains on code and instruction-following** tasks.
>
> |  | MATH | MBPP | xLAM | AlpacaEval |
> | --- | --- | --- | --- | --- |
> | ListRanker (ours) | **46.3** | **54.5** | **32.6** | **30.7** |
> | Self-Consistency | 44.9 | 41.9 | 24.6 | 20.4 |
> | First Sample | 25.1 | 41.9 | 10.6 | 20.4 |
>
> The second category includes methods such as Contrastive Decoding [3] and DoLa [4], which refine the model’s output distribution during generation. These approaches are **orthogonal to ours**, which focuses on post-sampling ranking. They are complementary to our method and baselines, and can be integrated independently; therefore, we do not include them as direct comparisons in our evaluation.
>
> [1] Chen,X., et al, RM-R1: Reward Modeling as Reasoning.
>
> [2] Wang, X., el al. Self-Consistency Improves Chain of Thought Reasoning in Language Models.
>
> [3] Li, X., et al. Contrastive Decoding: Open-ended Text Generation as Optimization.
>
> [4] Chuang, Y., et al. DoLa: Decoding by Contrasting Layers Improves Factuality in Large Language Models.
>
> ---
>
> **W2. Limited model coverage.**
>
> Thanks for your suggestions. To further validate the generality of our method, we additionally include results on **Gemma3-4B-it**, thereby expanding the coverage in both model scale and backbone diversity. The results on the MATH and the CODE task are shown in the table below.
>
> | Method | Parameter | MATH | CODE |
> | --- | --- | --- | --- |
> | ListRanker (ours) | **0.20M** | 72.2 | 51.4 |
> | PointRanker (ours) | **0.19M** | **72.4** | **51.7** |
> | RM (gpt2) | 137M | 69.2 | 50.3 |
> | RM (Qwen7B) | 161M/7.6B | 69.1 | 50.9 |
> | Beam Search | — | 67.4 | 49.2 |
> | First Sample | — | 63.5 | 48.8 |
>
> Similar to the results in our paper, both the listwise and the pointwise rankers **significantly improve model performance** on both the MATH task and the CODE task. These experiments furthur prove the effectiveness of our proposed ranker across different scales and backbone models.
>
> ---
>
> **W3. Narrow scope of transferability.**
>
> According to your suggestions, we conduct cross-domain transfer experiments using Llama similarly to the transfer experiements in the paper. We train the ranker on a single task type (MATH or CODE) and evaluate its generalization on the other task.
>
> |  | To MATH | To CODE |
> | --- | --- | --- |
> | Ranker From MATH | **46.3** | 51.2 (-3.3) |
> | Ranker From CODE | 43.4 (-2.9) | **54.5** |
> | GPT2 | 42.9 (-3.4) | 47.7 (-6.8) |
>
> The Table shows that rankers trained on any individual task maintain **robust performance on the other one**. Notably, the transfer performance is even better than the reward model using GPT2 (43.4 v.s. 42.9 on MATH and 51.2 v.s. 47.7 on CODE). This furthur highlights the system’s adaptability to unseen domains.
>
> Furthermore, unlike recent general reward models, our ranker is **highly efficient and easy to train**. Its lightweight design supports pairing **one base model with multiple task-specific rankers**, enabling flexible deployment with minimal overhead. This makes our Language System framework a practical and scalable solution for adapting to diverse downstream tasks under real-world resource constraints.
>
> ---
>
> **W4. Missing ablation on ranker scale.**
>
> Thank you for the insightful comment. We have already analyzed **ranker scaling** in **Table 5 (Sec. 4)**. Increasing Transformer blocks from 1 to 4 yields only marginal gains (e.g., 46.3 → 46.9 on MATH), while computational cost grows linearly. This supports our default use of a **single block** for optimal efficiency–performance trade-off.
>
> We also compare two architectural variants: (i) a **listwise** ranker with Transformer blocks and (ii) a **pointwise** ranker with MLP blocks. As shown in **Tables 2 and 5**, both perform well, suggesting that the primary benefit comes from **reusing base model features**, rather than increasing ranker complexity.
>
> Additionally, **Table 4** presents fine-grained ablations on the ranker architecture, verifying the contribution of each component. For example, removing **instruction features** (i.e., user intent) leads to a clear performance drop, highlighting the importance of modeling instruction–response alignment and further supports our recommender-system-inspired design perspective.
>
> ---
>
> **Q1.** The performance with 10–20 candidate responses.
>
> We believe there may be a misunderstanding. **At inference time, we only sample 10 candidate responses**, which aligns with realistic latency budgets. The setting with 100 candidates is used **only during ranker training** to construct a diverse and informative training set—this is a common practice and is also used when training reward models. All results reported in Tables 2–6 are obtained using **10 candidates during inference**, under which our ranker consistently **outperforms or matches** the performance of reward models.
>
> To further assess generality, we also evaluate the ranker’s performance across different numbers of candidate responses (from 1 to 100), as shown in **Figure 4**. The results indicate that performance generally improves as the number of samples increases, but our method remains effective even with small candidate sets.
>
> ---
>
> **Q2.** CPU benchmarks.
>
> To our knowledge, **there are no prior benchmarks on on-device adaptation for LLM ranking**, as previous approaches typically rely on large reward models that are impractical to train or deploy on CPU devices. In contrast, our ranker is the **first lightweight design (<0.5M parameters)** that can be efficiently trained and run on CPUs (see Table 3), making **on-device adaptation both feasible and practical**.
>
> ---
>
> **Q3.** Integrate sampling and ranking into a unified training loop.
>
> Thank you for this great suggestion. We conducted a preliminary experiment where the **ranker guides sampling in a loop**:
>
> 1. train a pointwise ranker on part of the data,
> 2. use it to select “hard” responses (misclassified ones) from new data,
> 3. retrain the ranker on these, and repeat 3 times.
>
> On **Qwen2.5‑7B‑Instruct (MATH)** this improves accuracy **from 75.2  to  76.4**. This shows that integrating sampling and ranking can further boost performance. Exploring this unified training paradigm and extensions such as data selection during pretraining is an exciting direction for future work.
>
> ---
>
> **L1.**   I/O overhead with large number of outputs (K=100).
>
> This concern appears to stem from a misunderstanding regarding the output number. As clarified in **Response to Q1**, our method uses only **10 candidate responses** during **inference**, not 100. With 10 candidates, the total size of hidden states is approximately **100KB**, resulting in **minimal I/O overhead**. Combined with the ranker’s lightweight architecture (<0.5M parameters), our method remains highly efficient and is well-suited for **real-time** or **low-resource** deployment scenarios.

---

> > ### Comment · Reviewer_L4gf · 2025-08-06
> >
> > Thank you for your efforts in addressing my concerns. Most of my concerns have been resolved, and I will keep my score unchanged.

---

> > > ### Author Response · Authors · 2025-08-08
> > >
> > > Thank you for your feedback! We are pleased to know that our responses resolved your concerns. If there are any further questions or suggestions, please don’t hesitate to let us know. Have a great day!

---

> ### Author Response · Authors · 2025-08-05
> **Request for Further Discussion**
>
> Dear Reviewer L4gf,
>
> We apologize for the interruption and would like to kindly request your further discussion and guidance regarding our work.
>
> In our response, we have provided detailed explanations to address your concerns. For example, we included **comparisons with GRM and self‑consistency** to further validate the advantages of our method, added results on **Gemma3‑4B‑it** to strengthen the generality of our conclusions, and reported **transfer performance between math and code tasks** to demonstrate broader transferability.
>
> We sincerely thank you again for your thorough review and constructive feedback, and we very much hope to engage in further discussion with you.
>
> Sincerely,
>
> The Authors

---

### Official Review · Reviewer_RWdC · 2025-07-04

**Clarity:** 3
**Significance:** 2
**Originality:** 2
**Rating:** 4
**Confidence:** 4

**Summary:**

This paper addresses the output decoding process of large language models from a novel perspective. It introduces an additional ranking module based on Transformer architecture to replace components like the reward module in RL, aiming to enhance the quality of final model outputs. Experimental results demonstrate that the proposed method achieves comparable performance to existing approaches while significantly reducing model size.

**Questions:**

Please refer to the weaknesses part.

**Ethical Concerns:**

["NO or VERY MINOR ethics concerns only"]

**Final Justification:**

I have revisited the paper and realized that I had misunderstood the application of the point-wise score, so I no longer have questions about W1. Additionally, I suggest placing the explanation of when to use each ranker at the end of Section 2, as this may help readers better understand the content. For W3, I agree that this approach can save resources while improving efficiency. After consideration, I will update my rating to "4: Borderline accept".

**Limitations:**

1. The model design and experimental setup are inconsistent with the stated motivation.

2. The experimental comparisons are insufficient, with inadequate validation across different scales and backbone models.

**Quality:**

2

**Strengths And Weaknesses:**

Strengths:
1. This paper offers a unique perspective by viewing final model output from the standpoint of recommender systems. This opens up new directions for optimizing language models from alternative angles.

2. This paper is well-organized and clearly written, making it easy to follow and understand.

3. Experimental results show that the proposed method achieves comparable performance with substantially reduced computational cost and model size.

Weaknesses:
1. Although the paper emphasizes the motivation of adopting a recommender system perspective, the actual implementation uses list-wise/point-wise prediction with a Transformer/MLP architecture. The authors are suggested to emphasize the task and scenario adaptation of the ranker at the end of Section 2.

2. Compared to methods that use GPT-2 or similar models to compute rewards, this approach adopts a structurally simpler method. However, the paper lacks validation on complex, large-scale tasks, making it difficult to determine whether the promising results are due to the simplicity of the tasks or limited data diversity, which might allow a small model to perform well.

3. The method performs well on smaller models (e.g., 7B), but its performance on larger models (e.g., 32B) is less convincing. Further experiments are needed to demonstrate the effectiveness of the approach at scale.

---

> ### Author Rebuttal · Authors · 2025-07-31
>
> We sincerely appreciate your recognition of our paper’s **writing quality**, the **novelty** of the recommender-system perspective, and the **strong performance achieved under limited resource constraints**. We believe that some of the concerns raised may arise from misunderstandings, which we clarify below with further explanations and additional experimental results.
>
> ---
>
> **W1.** Although the paper emphasizes the motivation of adopting a recommender system perspective, the actual implementation uses point-wise prediction with a Transformer architecture. As the authors themselves note in the experimental setup, the model essentially performs a binary classification task, functioning more like a rating predictor than a true ranker. By the way, what if two results are classified as positive?
>
> We would like to clarify two key points: **(1)** we do **not** use only point-wise prediction with a Transformer architecture, and **(2)** our ranker does **not** essentially perform a binary classification task.
>
> As described in Section 2 and illustrated in Figure 3, our framework includes **two distinct rankers** following common recommender system paradigms: a listwise ranker and a pointwise ranker [1-2]. Both rankers compute the similarity between the instruction feature and each candidate response feature. The **listwise ranker** processes all candidate responses jointly through a Transformer block, enabling it to model inter-response interactions and calculate similarity scores in a holistic manner. In contrast, the **pointwise ranker** computes the similarity between the instruction and each response independently via an MLP block and ranks the responses based on their predicted similarity scores. Therefore, it is inaccurate to characterize our method as solely point-wise.
>
> **During inference**, both rankers sort the candidate responses based on their similarity scores and **select the one with the highest score**. As shown in Section 2, we design task-specific training objectives to ensure that the similarity scores effectively reflect response quality. For example:
>
> - For **math**, **code**, and **tool-use** tasks, where responses can be clearly labeled as correct or incorrect, we apply a classification loss that increases the scores of correct responses and decreases those of incorrect ones.
> - For general **instruction-following tasks** (Appendix B), where no absolute correctness labels are available and only coarse-grained user ratings exist, we use a regression loss to fit the observed ratings directly.
>
> Therefore, the binary classification is merely a proxy training objective used on certain tasks to help the ranker learn meaningful similarity scores—this is a common and well-established practice in recommender systems. The concern about “two results being classified as positive” does not apply. Our system does not rely on hard classification; instead, it ranks all responses based on their real-valued similarity scores.
>
> [1] Zehlike, M., et al. Fairness in ranking, part ii: Learning-to-rank and recommender systems.
>
> [2] Liu, W., et al. Neural re-ranking in multi-stage recommender systems: A review.
>
> ---
>
> **W2.** Compared to methods that use GPT-2 or similar models to compute rewards, this approach adopts a structurally simpler method. However, the paper lacks validation on complex, large-scale tasks, making it difficult to determine whether the promising results are due to the simplicity of the tasks or limited data diversity, which might allow a small model to perform well.
>
> We respectfully disagree with this comment and believe that our evaluation setup is sufficient to support the conclusions. As noted by reviewers **L4gf** and **gdjF**, the **strong empirical results across various tasks** is one of the **strengths** of our work.
>
> **Regarding data diversity**, we evaluate four representative task categories: **math**, **code**, **tool use**, and **instruction following**. The first three reflect core LLM capabilities of broad interest [6-8], while the last assesses general instruction-following ability. In particular, the instruction-following task leverages **AlpacaEval** [1], a large-scale benchmark that aggregates diverse instructions from sources such as Self-Instruct [2], OASST, Anthropic’s Helpful dataset [3], Vicuna [4], and Koala [5], offering a comprehensive and challenging test of general LLM capabilities.
>
> **Regarding benchmark complexity and credibility**, all benchmarks used in our experiments are widely adopted and authoritative within their respective domains:
>
> - **MATH** and **MBPP** are standard benchmarks for mathematical reasoning and code generation. They are featured in nearly all major LLM technical reports [6–8].
> - **Tool use** is evaluated using **xLAM**, a representative dataset in tool-augmented language modeling that involves large-scale tool library with over 3,600 tools [9].
> - **Instruction following** is evaluated using **AlpacaEval**, which is specifically designed to test general-purpose language understanding and generation, and is also frequently reported in major LLM technical reports [6].
>
> These benchmarks provide a broad and credible testbed for evaluating models across both narrow and general tasks, and we believe they sufficiently validate the effectiveness of our approach.
>
> The strong performance of our ranker with only <5M parameters across diverse and challenging tasks is not due to task simplicity, but rather a result of its efficient design: it effectively leverages the rich features already extracted by the base models, enabling effective ranking with minimal additional overhead.
>
> [1] Dubois, Y., et al. AlpacaFarm: A Simulation Framework for Methods that Learn from Human Feedback.
>
> [2] Wang, Y., et al. Self-instruct: Aligning language models with self-generated instructions.
>
> [3] Bai, Y., et al. Training a helpful and harmless assistant with reinforcement learning from human feedback.
>
> [4] Chiang, W., et al. Vicuna: An open-source chatbot impressing gpt-4 with 90% chatgpt quality.
>
> [5] Geng, X., et al. Koala: A dialogue model for academic research, March.
>
> [6] DeepSeek-AI. DeepSeek-V3 Technical Report.
>
> [7] Gemini Team, Google. Gemini 2.5: Pushing the Frontier with Advanced Reasoning, Multimodality, Long Context, and Next Generation Agentic Capabilities.
>
> [8] Qwen Team, Qwen3 Technical Report.
>
> [9] Liu, Z., et al. APIGen: Automated PIpeline for Generating Verifiable and Diverse Function-Calling Datasets.
>
> ---
>
> **W3.** The method performs well on smaller models (e.g., 7B), but its performance on larger models (e.g., 32B) is less convincing. Further experiments are needed to demonstrate the effectiveness of the approach at scale.
>
> We would like to clarify that the performance of our method on the 32B model is also strong, for the following reasons:
>
> 1. **The average accuracy is competible to that of a trained 32B reward model (76.1% vs. 76.7%)**, demonstrating that our approach remains effective even at large scale.
> 2. **Our method introduces only ~0.35M additional parameters during both training and inference.** The ranker is lightweight enough to be trained and deployed even on CPU. In contrast, using a separate 32B reward model requires training an additional large model and computing reward scores for each response at inference stage—resulting in significant time and computational overhead that limits practical usability.
>
> The strength of our approach lies not only in its **effectiveness**, but also in its **efficiency**. From 7B to 32B base models, our method achieves consistently strong performance with similarly lightweight rankers (0.30M → 0.35M). This demonstrates its scalability and general applicability across model sizes.
>
> Moreover, the low overhead of our ranker brings many benefits. For example, a single base model can be paired with different rankers to enable personalized adaptation for diverse user needs simultaneously, with minimal additional overhead.
>
> ---
>
> **L1.** The model design and experimental setup are inconsistent with the stated motivation.
>
> We would like to clarify that our model design and experimental setup are aligned with the stated motivation. This concern may stem from a misunderstanding of our approach, which we have clarified in detail in our **response to W1.**
>
> ---
>
> **L2.** The experimental comparisons are insufficient, with inadequate validation across different scales and backbone models.
>
> Thank you for your suggestion. We have conducted experiments across a wide range of tasks using both 7B and 32B base models, covering two prevalent architectures: LLaMA and Qwen. To further validate the generality of our method, we additionally include results on **Gemma3-4B-it**, thereby expanding the coverage in both model scale and backbone diversity. The results on the MATH and the CODE task are shown in the table below.
>
> | Method | Parameter | MATH | CODE |
> | --- | --- | --- | --- |
> | ListRanker (ours) | **0.20M** | 72.2 | 51.4 |
> | PointRanker (ours) | **0.19M** | **72.4** | **51.7** |
> | RM (gpt2) | 137M | 69.2 | 50.3 |
> | RM (Qwen7B) | 161M/7.6B | 69.1 | 50.9 |
> | Beam Search | — | 67.4 | 49.2 |
> | First Sample | — | 63.5 | 48.8 |
>
> Similar to the results in our paper, both the listwise and the pointwise rankers significantly improve model performance on both the MATH task and the CODE task with Gemma3-4B-it. These experiments furthur prove the effectiveness of our proposed ranker across different scales and backbone models.

---

> ### Author Response · Authors · 2025-08-05
> **Request for Further Discussion**
>
> Dear Reviewer RWdc,
>
> We apologize for the interruption and would like to kindly request further discussion and guidance regarding our work.
>
> In our response, we have provided detailed explanations to address your concerns. For example, we clarified the misunderstandings regarding our method’s ranker structure and training settings, and we we added **Gemma3‑4B‑it** results to validate our approach on a broader range of model scales and backbones.
>
> We sincerely thank you again for your thorough review and constructive feedback, and we very much hope to engage in further discussion with you.
>
> Sincerely,
>
> The Authors

---

> ### Author Response · Authors · 2025-08-07
>
> Dear Reviewer RWdc,
>
> We apologize for the interruption. As the discussion deadline is approaching, we would like to kindly follow up to see if you might have any further thoughts on our response. We would greatly appreciate your feedback.
>
> Sincerely,
> The Authors

---

> > ### Comment · Reviewer_RWdC · 2025-08-08
> >
> > Thanks for your detailed reply. I have revisited the paper and realized that I had misunderstood the application of the point-wise score, so I no longer have questions about W1. Additionally, I suggest placing the explanation of when to use each ranker at the end of Section 2, as this may help readers better understand the content. For W3, I agree that this approach can save resources while improving efficiency. After consideration, I will update my rating to "4: Borderline accept"

---

> > > ### Author Response · Authors · 2025-08-08
> > >
> > > Thank you for raising the score and for your constructive suggestion. We’re glad our responses addressed your concerns, and we will place the explanation at the end of Section 2. If you have any additional questions or feedback, please feel free to let us know. Wishing you a wonderful day!

---

### Comment · Area_Chair_J5Xw · 2025-08-02

Dear Reviewers,

Thank you for your time and effort in reviewing this manuscript. The authors have submitted their rebuttal, and we would greatly appreciate it if you could review their responses at your earliest convenience.

If you have any further questions or concerns regarding the rebuttal, please don't hesitate to discuss. Thanks for your contribution to NeurIPS 2025.

Best regards,
AC

---

### Note · Authors · 2025-08-13

Dear Area Chair and Reviewers,

We sincerely thank you for your efforts and patience in our paper reviewing process. We are pleased that most concerns raised by Reviewers RWdC and L4gf have been addressed. As Reviewers gdjF and pdDQ were not available for the discussion, we lack clarity on their perspectives. Below, we summarize the rebuttal discussions and present our final remarks.


- **Good Experimental Results.** After discussion and additional experiments, most reviewers agreed that the **experiments are comprehensive** and that our method demonstrates **strong performance**. This suggests that Language System is a novel, effective, and efficient framework for enhancing LLM capabilities across general domains.

- **Detailed Analysis and General Transferability**. Most reviewers acknowledged that we conducted **detailed analytical experiments**, particularly on **ranker scale and candidate number (K)**. Besides, we provided **cross-domain transfer results** in response to Reviewer L4gf, further validating the general transferability of our method. These results indicate that our approach achieves an **optimal balance between effectiveness and efficiency**, while also exhibiting **strong transferability**.

- **Deep Theoretical Analysis**. In response to Reviewer gdjF, we added an **deep theoretical analysis** to explain why the two-stage Language System, inspired by recommender systems, offers **unique advantages** beyond empirical results.

- **Novelty.** We clarify that our work introduces several pioneering elements: (1) we are the **first to explicitly repurpose LLM hidden states for efficient response reranking** during inference; (2) drawing **inspiration from recommender systems**, we propose an **instruction–response feature alignment mechanism** (validated through the ablation study in Table 4); and (3) we establish the **first lightweight ranking framework enabling personalized LLM systems** through multiple adaptable rankers, overcoming resource constraints unresolved by prior work. As highlighted by **Reviewers RWdC and L4gf**, novelty is one of the **key strengths** of our work.

---

Thank you once again for your support!

Best regards,

Authors

---

### Decision · Program_Chairs · 2025-09-17

**Decision:**

Accept (poster)

**Comment:**

**Summary:**
This paper introduces "Language System", a novel framework that re-frames LLM decoding as a recommender system task, using a lightweight ranker to rerank candidate responses. The method demonstrates strong, efficient performance across diverse tasks (math, code, tool use) with minimal parameter overhead (<0.5M). The authors provided a comprehensive rebuttal, adding cross-domain transfer results to address reviewer concerns.

**Pros:**
*  Fresh, well-motivated recommender-system analogy for LLM output selection.
*  Achieves performance comparable to large reward models with a tiny ranker.
*  Comprehensive experiments across multiple domains and model families (LLaMA, Qwen, Gemma) support the claims.

**Cons:**
*   While a theoretical analysis was added, its depth and direct connection to the empirical success remain a point for future work.
*   Scaling laws for the ranker itself are not fully explored.
*   The comparison, though expanded, could still be more exhaustive against the full spectrum of modern decoding-time optimization techniques, such as some confidence-aware inference scaling methods.

Overall, I like the idea of this paper. However, the paper writing could be further optimized. For example, the abstract should clearly state what key conclusions were drawn from the analogy between LLM decoding and RecSys, which led to the identification of limitations. The introduction should explain how the ranker is optimized, as well as address the generalizability of the ranker.